# Novel nucleoside analogs exhibit potent intracellular and *in vivo* activities against *Mycobacterium avium*

Ho-Sung Park,[1,2,3,4] Do Young Kim,[5] Ji Seok Oh,[5] Dong Hyon Koo,[5] Suyeon Yeom,[5] Seungwoo Kim,[5] Arjun Gontala,[5] Sang-Yeop Lee,[6] Dong Ho Kim,[1,2] Kyungho Woo,[1,2] Seung Il Kim,[6] Jun Young Heo,[1,2,4,7] Woosuk Chung,[1,2,4,8,9] Hak Joong Kim,[5] Chul Hee Choi[1,2,3,4]

**ABSTRACT** *Mycobacterium avium* is a major causative agent of nontuberculous mycobacterial pulmonary disease, which poses therapeutic challenges owing to its intrinsic drug resistance and the need for prolonged multidrug regimens. In this study, we identified two novel nucleoside analogs, MCCB-04-35 and MCCB-04-37, as potential therapeutic candidates against *M. avium* infection. Both compounds exhibited significant bacteriostatic activity *in vitro* and in infected macrophages, with minimal cytotoxicity. Time–kill kinetics and MIC assays confirmed their potent inhibitory effects, particularly against slow-growing mycobacteria. Checkerboard synergy testing revealed additive to synergistic interactions with clinically used antibiotics such as clarithromycin and ciprofloxacin. In a mouse model of chronic lung infection, both compounds significantly reduced pulmonary bacterial burden, inflammatory cytokine levels, and histopathological damage. Transcriptomic analysis of treated *M. avium* revealed the downregulation of key metabolic pathways, including oxidative phosphorylation and nitrogen metabolism, indicating disruption of intracellular energy homeostasis. These findings suggest that MCCB-04-35 and MCCB-04-37 exert antimicrobial effects through metabolic interference and may serve as effective therapeutic agents either alone or in combination for treating *M. avium* infections.

**IMPORTANCE** Pulmonary disease caused by *Mycobacterium avium complex* (MAC) is notoriously difficult to treat due to intrinsic antibiotic resistance and the need for prolonged multidrug therapy, often poorly tolerated with suboptimal outcomes. The identification of new therapeutic candidates with novel mechanisms of action is urgently needed. Here, we report two novel nucleoside analogs, MCCB-04-35 and MCCB-04-37, exhibiting strong anti-mycobacterial activity against *M. avium* both *in vitro* and *in vivo*, with minimal cytotoxicity. These compounds showed additive to synergistic effects when combined with existing antibiotics such as clarithromycin. In a mouse model of chronic lung infection, they significantly reduced bacterial burden, inflammation, and tissue damage. Transcriptomic profiling revealed downregulation of metabolic pathways essential for bacterial energy production, suggesting a unique mechanism of antimicrobial action. Our findings provide promising leads for the development of more effective treatments for MAC pulmonary disease, either as monotherapy or in combination with current drugs.

**KEYWORDS** *Mycobacterium avium*, nucleoside analogs, antimicrobial resistance, intracellular survival, synergistic therapy

**Peer Reviewers** Erik Yukl, New Mexico State University, Las Cruces, New Mexico, USA; Jichan Jang, Gyeongsang National University, Gyeongsangnam-do, Republic of Korea

Address correspondence to Hak Joong Kim, hakkim@korea.ac.kr, or Chul Hee Choi, choich@cnu.ac.kr.

The authors declare no conflict of interest.

*[This article was published on 23 January 2026 with an error in author Seungwoo Kim's name. The error was updated in the current version, posted on 28 January 2026.]*

Nontuberculous mycobacteria (NTM) are opportunistic pathogens that are widely distributed in soil and water and can cause pulmonary and extrapulmonary infections, particularly in individuals with underlying lung diseases, such as bronchiectasis and chronic obstructive pulmonary disease (1, 2). Among them, the *Mycobacterium*

*avium* complex (MAC), comprising *M. avium* and *Mycobacterium intracellulare*, is the most prevalent cause of NTM pulmonary disease worldwide (3, 4). MAC-induced pulmonary disease is a chronic and slowly progressive condition that leads to irreversible lung damage through persistent inflammation, including bronchiectasis, cavitary lesion formation, and decline in pulmonary function (5, 6). Although clinical outcomes vary among patients, many experience deterioration in lung function, decreased quality of life, and increased mortality risk (7).

Treatment of MAC-induced pulmonary disease remains challenging owing to the intrinsic resistance of MAC organisms and the requirement for long-term multidrug therapy. Standard treatment involves a combination of macrolides (clarithromycin or azithromycin), rifampin, and ethambutol administered for at least 12 months after sputum culture conversion (8). However, clinical outcomes remain suboptimal, with culture conversion rates ranging between 32% and 65% and frequent relapse following treatment cessation (9, 10). Furthermore, prolonged antibiotic therapy is associated with adverse effects that can compromise adherence and contribute to developing drug resistance (11).

To address the limitations of current regimens, recent studies have focused on metabolic processes that are critical for the intracellular survival of *M. avium* complex (MAC), particularly oxidative phosphorylation and nitrogen assimilation (12, 13). Although these pathways are broadly conserved among bacteria, several reports have demonstrated that MAC becomes highly dependent on them within host macrophages, rendering these processes conditionally essential under hypoxic and nutrient-limited conditions. Nucleoside analogs can disrupt intracellular nucleotide pools and nucleic acid metabolism, thereby imposing metabolic stress that may secondarily affect energy production and nitrogen assimilation. Such host-adaptive metabolic pathways represent promising therapeutic targets for impairing MAC persistence and enhancing treatment efficacy (14).

In this study, we aimed to develop novel drugs against *M. avium*. We synthesized and evaluated two novel nucleoside analogs, MCCB-04-35 and MCCB-04-37, for their bactericidal effects. Additionally, we aimed to elucidate their mechanisms of action, focusing on their impact on bacterial metabolic pathways.

## RESULTS

### Chemical synthesis

MCCB-04-35 and MCCB-04-37 were synthesized from 2-bromobenzaldehyde derivatives (**1a/1b**). First, the aldehyde group of compounds **1a** and 1**b** was protected as an acetal, followed by Suzuki–Miyaura cross-coupling with potassium alkenyltrifluoroborate to introduce a vinyl group (15). The resulting aromatic vinyl acetal (**2a/2b**) then underwent Sharpless asymmetric dihydroxylation to selectively yield the *R*-isomer of the aromatic diol acetal (16). To prevent side reactions during subsequent cyclization, the primary alcohol was selectively protected with a pivaloyl group, affording compounds **3a/3b**. Notably, cyclization without this protection led to undesired six-membered ring byproducts (17). The exposure of **3a/3b** to acidic conditions removed the acetal protecting group, triggering cyclization to form hemiacetal intermediates. For nucleobase introduction, a typical Vorbrüggen glycosylation protocol was used (18). Specifically, each cyclic hemiacetal intermediate was reacted with *in-situ* prepared *O,O*-bis(trimethylsilyl)−5-fluorouracil in the presence of Sn(IV) chloride as a Lewis acid, promoting the formation of *N*-glycosidic bond. The resulting approximately 2:1 mixture of 1′*R* and 1′*S* stereoisomers was separated *via* preparative reverse-phase HPLC. Finally, the pivaloyl protecting group was removed under saponification conditions, yielding the target compounds, MCCB-04-35 and MCCB-04-37 (Fig. 1). The stereochemistry of MCCB-04-37 was assigned by the absence of nuclear Overhauser effect signals between 1′- and 4′-protons, in contrast to their corresponding 1′,4′-*syn* diastereomers, which exhibited clear nuclear Overhauser effect correlations (Fig. S1 and S2). In addition, distinct differences in chemical shift at the 1′-, 5′-, and 6-positions were observed

**FIG 1** Synthesis of MCCB-04-35 and MCCB-04-37.

between MCCB-04-37 and its diastereomer. These characteristic patterns of chemical shift were used to support the assignment of an *anti*-stereochemical relationship between the 1′- and 4′-positions in MCCB-04-35, compared with its stereoisomer.

### *In vitro* antimicrobial activity of MCCB-04-35 and MCCB-04-37 against *M. avium*

The minimum inhibitory concentrations (MICs) of MCCB-04-35 and MCCB-04-37 against various mycobacterial strains were determined using the broth microdilution method in accordance with CLSI guidelines. MCCB-04-35 exhibited MICs of 20 and 5 µg/mL against *M. avium* MAH 104 reference strain and *M. intracellulare*, respectively. In contrast, MCCB-04-37 showed MICs of 40 and 20 µg/mL against clinical isolates of *M. avium* and *M. intracellulare*, respectively. Notably, for both non-clinical and clinical strains of *M. abscessus*, MCCB-04-37 exhibited an MIC of 10 µg/mL. The detailed MIC values for each strain are summarized in Table S1. The tested strains were categorized as rapid- or slow-growing mycobacteria based on their growth rates. MCCB-04-35 and MCCB-04-37 exhibited bactericidal activities against *M. avium* and *M. intracellulare*, with stronger inhibitory effects observed particularly against slow-growing mycobacteria. To determine the optimal inhibitory concentrations of the compounds against *M. avium*, dose-response curves were generated using a concentration range of 0.1–500 µg/mL, and half-maximal inhibitory concentration ($IC_{50}$) values were calculated (Fig. 2A). The $IC_{50}$ values of MCCB-04-35 and MCCB-04-37 were 1.3 and 1.36 µg/mL, respectively (Fig. 2B). To evaluate the time-dependent antibacterial activity of MCCB-04-35 and MCCB-04-37, time-kill kinetics and optical-density ($OD_{600}$) growth-curve analyses were performed (Fig. 2C through F). Both compounds exhibited clear concentration-dependent effects against *M. avium*. At concentrations ≥100 µg/mL, viable colony forming unit (CFU) counts decreased to undetectable levels within 5 days, demonstrating bactericidal activity. At 10 µg/mL, CFU values remained stable for approximately 20 days, consistent with a bacteriostatic effect, whereas 1 µg/mL caused an approximate 6-day delay in bacterial growth relative to the untreated control. Clarithromycin also showed a concentration-dependent response, displaying a bacteriostatic pattern at 0.25 µg/mL but exhibiting bactericidal activity at 10 µg/mL, where CFU counts markedly decreased during the observation period. Consistent with the CFU data, $OD_{600}$ measurements revealed a corresponding reduction in bacterial growth in response to both compounds.

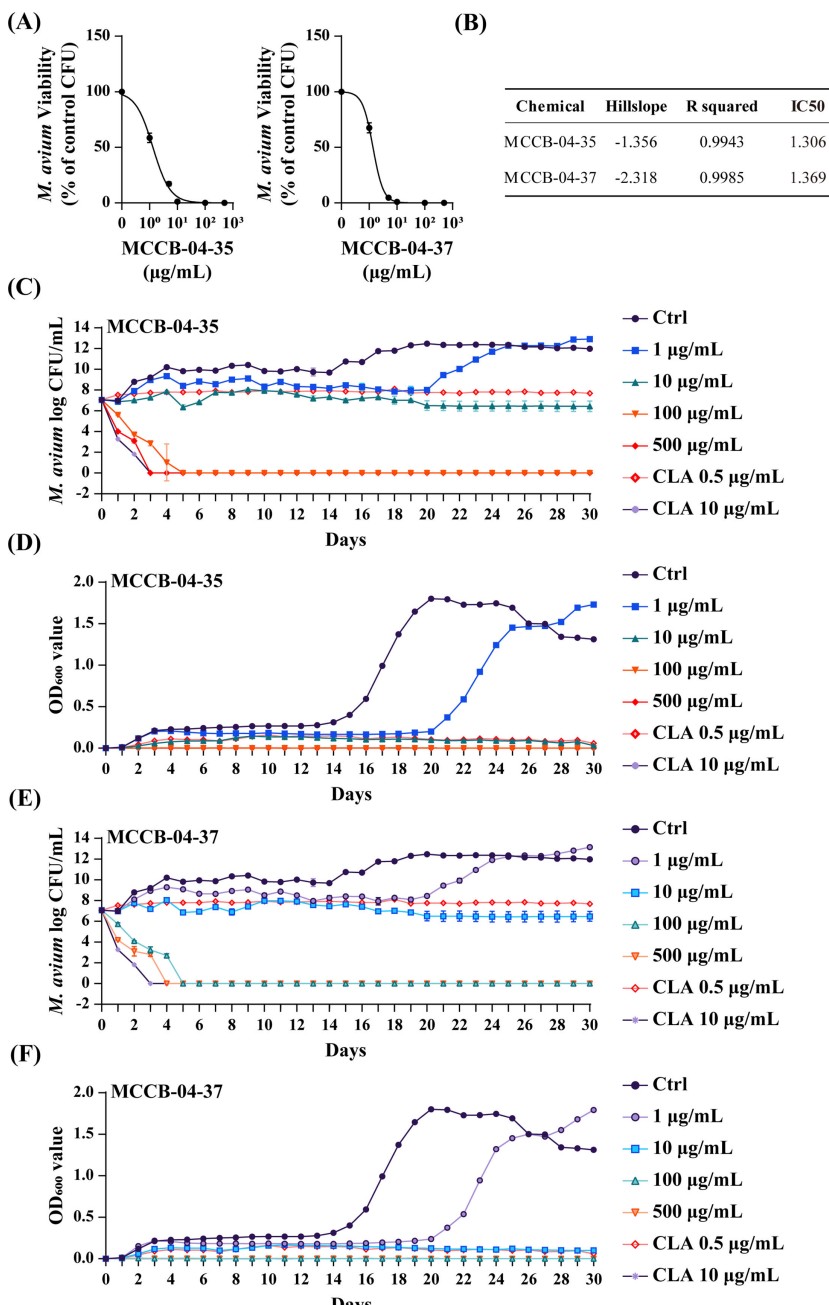

**FIG 2** Activities of MCCB-04-35 and MCCB-04-37 against *M. avium in vitro*. (A) Dose-response curves illustrating the half-maximal inhibitory effects of MCCB-04-35 and MCCB-04-37 on *M. avium* growth. The curves show the relationship between the log concentration of each compound (μg/mL) and the number of viable *M. avium* cells (CFU/mL). (B) $IC_{50}$ values (μg/mL) of MCCB-04-35 and MCCB-04-37 against *M. avium*. The "Hillslope" and "R squared" values represent the slope and goodness of fit of the dose–response curves, respectively. (C and D) Time–kill kinetics of MCCB-04-35 against *M. avium*. (C) $Log_{10}$ CFU/mL values of *M. avium* measured over 30 days following treatment. (D) Corresponding optical-density ($OD_{600}$) values indicating bacterial growth inhibition. (E and F) Time–kill kinetics of MCCB-04-37 against *M. avium*. (E) $Log_{10}$ CFU/mL values measured over 30 days. (F) Corresponding $OD_{600}$ values indicating bacterial growth inhibition. "ctrl" denotes the untreated control, and "DMSO" represents the solvent control.

TABLE 1 Synergistic activity of MCCB-04-35 and MCCB-04-37 in combination with conventional antibiotics

| Antibiotics | MIC (µg/mL) | | | FICI | Result |
|---|---|---|---|---|---|
| | Antibiotics single use | MCCB-04-35 single use | Combination therapy (antibiotics/35) | | |
| Amikacin | 0.5 | 50 | 0.25/25 | 1 | Additive |
| Ciprofloxacin | 4 | 50 | 1/12.5 | 0.5 | Synergism |
| Moxifloxacin | 8 | 50 | 1/25 | 0.625 | Additive |
| Linezolid | 4 | 50 | 0.5/25 | 0.625 | Additive |
| Clarithromycin | 0.25 | 25 | 0.06/12.5 | 0.75 | Additive |
| Antibiotics | MIC (µg/mL) | | | FICI | Result |
| | Antibiotics single use | MCCB-04-37 single use | Combination therapy (antibiotics/37) | | |
| Amikacin | 0.5 | 50 | 0.25/25 | 1 | Additive |
| Ciprofloxacin | 4 | 50 | 0.5/25 | 0.625 | Additive |
| Moxifloxacin | 8 | 50 | 1/25 | 0.625 | Additive |
| Linezolid | 4 | 50 | 1/25 | 0.75 | Additive |
| Clarithromycin | 0.25 | 50 | 0.06/12.5 | 0.5 | Synergism |

## Synergistic effects of MCCB-04-35 and MCCB-04-37 in combination with conventional antibiotics

Treating NTM infections is challenging owing to their high potential for developing resistance, often necessitating combination therapy. To evaluate the synergistic effects of MCCB-04-35 and MCCB-04-37 with clinically used antibiotics, a checkerboard assay was performed. MCCB-04-35 exhibited the weakest interaction with amikacin (fractional inhibitory concentration index, FICI = 1.0), indicating no synergistic effect (Table 1). In contrast, clarithromycin, moxifloxacin, and linezolid showed additive effects, with FICI values ranging between 0.625 and 0.75. Among the tested antibiotics, ciprofloxacin showed the most pronounced interaction with MCCB-04-35 (FICI = 0.5), representing the lowest FICI value among all tested combinations. The MIC of ciprofloxacin against *M. avium* was 4 µg/mL; however, it decreased fourfold to 1 µg/mL when combined with MCCB-04-35. Similarly, MCCB-04-37 showed no interaction with amikacin (FICI = 1.0), whereas additive effects were observed with ciprofloxacin, moxifloxacin, and linezolid (FICI = 0.625–0.75). Notably, clarithromycin exhibited a synergistic interaction with MCCB-04-37, with an FICI of 0.5. The MIC of clarithromycin was 0.25 µg/mL; however, it reduced fourfold to 0.06 µg/mL in combination with MCCB-04-37 (Table 1). To validate these checkerboard findings, CFU counts confirmed that ciprofloxacin combined with MCCB-04-35 (6.40 $\log_{10}$ CFU/mL) and clarithromycin combined with MCCB-04-37 (7.13 $\log_{10}$ CFU/mL) significantly reduced bacterial counts compared with either monotherapy, showing approximately 1.8–2.0 $\log_{10}$ (60- to 100-fold) lower CFU values (Fig. S3). Together, these findings demonstrate that MCCB-04-35 and MCCB-04-37 enhance the efficacy of specific antibiotics, particularly ciprofloxacin and clarithromycin, highlighting their potential utility in combination therapy for treating *M. avium* infections.

## Anti-*M. avium* effects of MCCB-04-35 and MCCB-04-37 in *M. avium*-infected macrophages

Both the test compounds at 500 µg/mL concentrations showed >20% cytotoxicity in AML12 cells; however, no significant cytotoxicity was detected at concentrations of 1–100 µg/mL (Fig. 3A). Therefore, bone marrow-derived macrophages (BMDMs) were infected with *M. avium* and treated with 40 µg/mL MCCB-04-35 or MCCB-04-37, followed by evaluation of intracellular and extracellular bacterial survival. MCCB-04-35 resulted in a 48.9% reduction in intracellular bacterial survival, whereas MCCB-04-37 reduced intracellular survival by 33.3%, both relative to the non-treated (NT) group. These

**(A)**

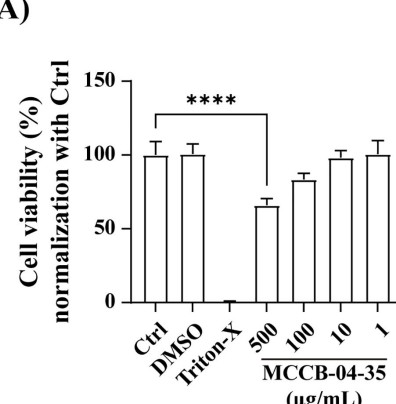
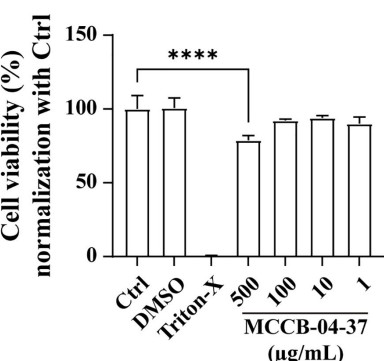

**(B)**

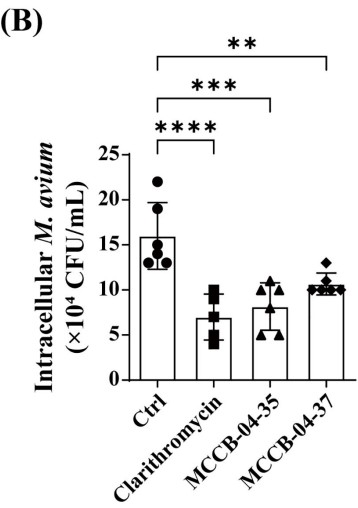

**(C)**

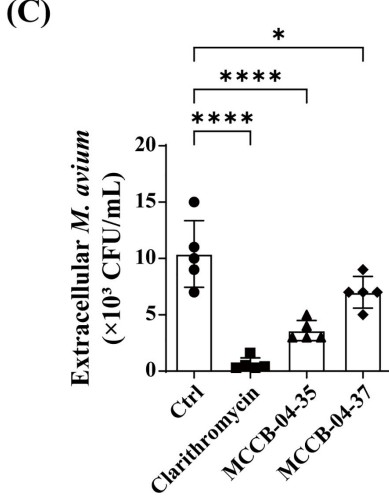

FIG 3 Effects of MCCB-04-35 and MCCB-04-37 on host cell viability and *M. avium* survival. (A) Cytotoxicity of MCCB-04-35 and MCCB-04-37 in AML12 cells. Data are presented as the percentage of viable cells relative to those of the untreated control. (B) Intracellular survival of *M. avium* in BMDMs following treatment with MCCB-04-35 and MCCB-04-37. The number of viable intracellular *M. avium* cells (CFU/mL) was quantified 24-h post-treatment. (C) Extracellular survival of *M. avium* in BMDMs following treatment with MCCB-04-35 and MCCB-04-37. The number of viable intracellular *M. avium* cells (CFU/mL) was quantified 24 h after treatment. Data are presented as mean ± SD from independent experiments. *$P < 0.05$, **$P < 0.01$, ***$P < 0.001$, and ****$P < 0.0001$ (one-way ANOVA).

reductions were comparable to the effect of clarithromycin (56.25%) used as a positive control (Fig. 3B). In addition, extracellular bacterial counts were significantly reduced, with MCCB-04-35 and MCCB-04-37 decreasing survival by 65.3% and 32.6%, respectively, relative to the NT group (Fig. 3C). Therefore, MCCB-04-35 and MCCB-04-37 inhibit the growth of *M. avium* and effectively suppress its intracellular persistence.

## Differentially expressed genes in MCCB-04-35- and MCCB-04-37-treated groups

In the MCCB-04-35-treated group, 137 genes were differentially expressed compared to those in the untreated control, including 67 upregulated and 70 downregulated genes (Fig. 4A). MCCB-04-37 treatment led to 129 DEGs, with 54 upregulated and 75 downregulated genes (Fig. 4B). Functional enrichment analysis of the DEGs (Fig. 4C and

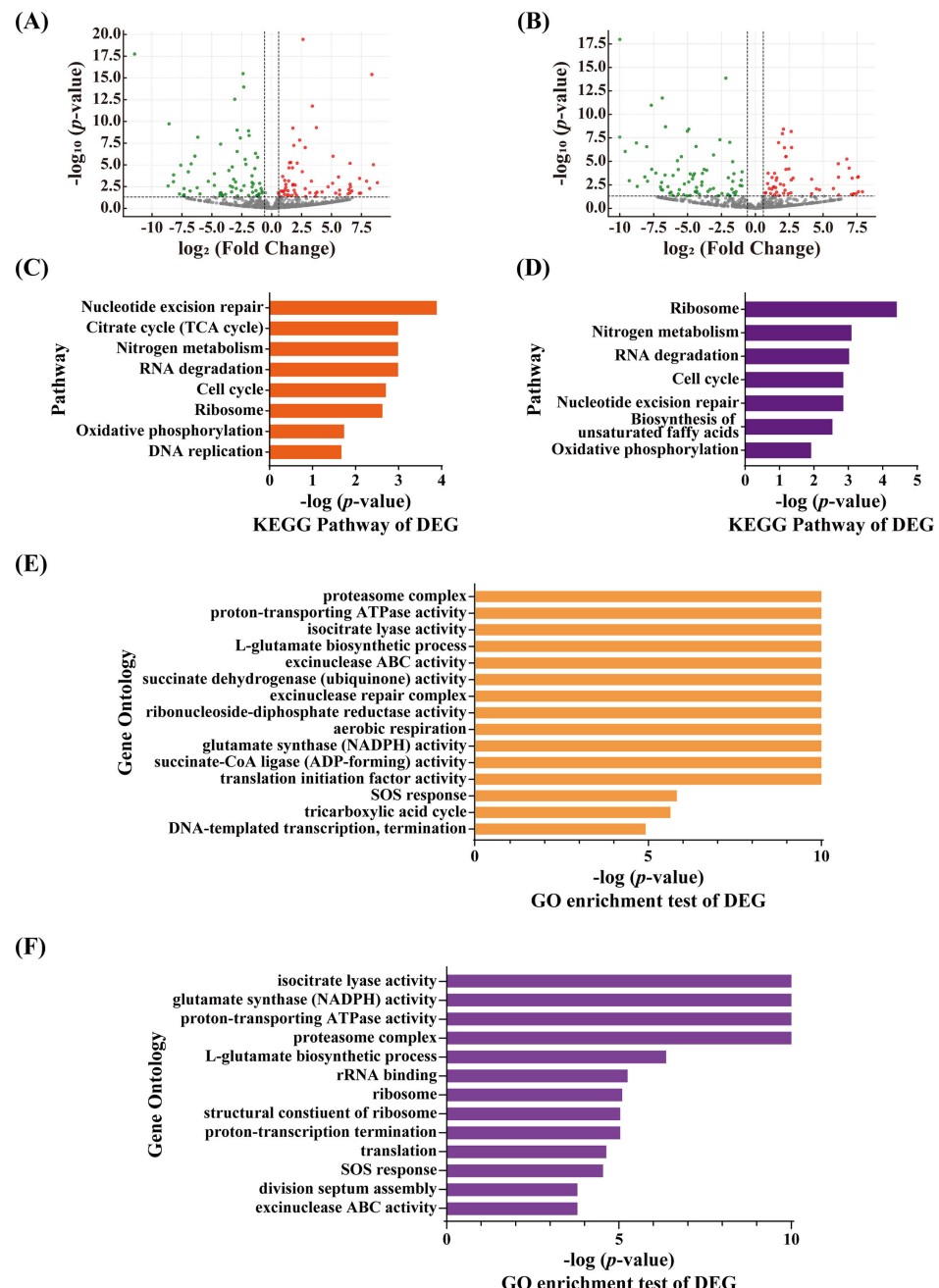

**FIG 4** Transcriptomic analysis of *M. avium* treated with MCCB-04-35 and MCCB-04-37. Volcano plots showing DEGs in *M. avium* treated with MCCB-04-35 (A) and MCCB-04-37 (B), compared to the genes in the untreated control. Red and green dots represent significantly upregulated and downregulated genes, respectively. Kyoto Encyclopedia of Genes and Genomes (KEGG) pathway enrichment analysis of DEGs from MCCB-04-35 (C) and MCCB-04-37 (D) treatment groups. Gene Ontology (GO) enrichment analysis of DEGs from MCCB-04-35 (E) and MCCB-04-37 (F) treatment groups. The data represent the results of independent RNA-seq experiments. Statistical significance was determined based on a hypergeometric test with $P < 0.05$.

D) indicated that both compounds elicited similar responses in pathways associated with drug-induced stress and bacterial survival. Genes related to nucleotide excision repair and RNA degradation, hallmarks of the drug stress response, were commonly upregulated. Conversely, metabolic pathways such as nitrogen metabolism and oxidative phosphorylation were significantly downregulated. Gene Ontology (GO) enrichment analysis further supported these findings (Fig. 4E and F). Notably, genes involved in

the L-glutamate biosynthetic process and glutamate synthase (NADPH) activity, key components of nitrogen metabolism, were downregulated. Additionally, genes encoding components of the proton-transporting ATPase and ATP synthase complexes, which are central to oxidative phosphorylation and ATP production, were suppressed. Specifically, *atpA* (MAV_RS07285), *atpG* (MAV_RS07290), and *atpD* (MAV_RS07295), encoding the $F_0F_1$ ATP synthase alpha, gamma, and beta subunits, respectively, were significantly downregulated during treatment. Representative DEGs corresponding to the enriched Kyoto Encyclopedia of Genes and Genomes (KEGG) pathways and GO terms described above, including their fold-change values and adjusted *P*-values, are summarized in Table S2. The complete list of all DEGs identified in the transcriptomic analysis, along with their gene names, functional annotations, accession numbers, $\log_2$ fold-change values, and adjusted *P*-values, is provided separately in Supplementary data_full list_GO analysis.xlsx at https://doi.org/10.6084/m9.figshare.30426742.v2.

## *In vivo* efficacy of MCCB-04-35 and MCCB-04-37 in a mouse model of chronic lung infection

The body weight, pulmonary bacterial burden, inflammatory cytokine levels, and histopathological changes of mice were assessed (Fig. 5A). Body-weight monitoring revealed no significant differences between the treatment groups, indicating no apparent drug-related toxicity (Fig. 5B). In the MCCB-04-35-treated group, pulmonary bacterial loads were significantly reduced compared to that in the infection control group, with a $\log_{10}$ reduction of 0.4 (60.7%) 1 week after treatment. Tumor necrosis factor alpha (TNF-α) and interferon gamma (IFN-γ) levels were decreased by 66.2% and 38%, respectively. At 2-week post-treatment, bacterial burden declined further by 75.1%, with TNF-α and IFN-γ levels reduced by 76.6% and 51%, respectively (Fig. S4). By 4 weeks, clarithromycin (CLA) treatment reduced pulmonary *M. avium* counts by 96.5% relative to the infection control, confirming its expected efficacy as a standard reference. Under the same conditions, MCCB-04-35 treatment showed a comparable reduction of 94.8%, demonstrating potent *in vivo* antibacterial activity (Fig. 5C). In bronchoalveolar lavage fluid (BALF), TNF-α and IFN-γ levels were 7.37 and 11.23 pg/mL, reflecting reductions by 81.6% and 81% compared to those in the infection control group (TNF-α, 40 pg/mL; IFN-γ, 59 pg/mL), respectively (Fig. 5D and E). Similarly, MCCB-04-37 treatment led to a 33.3% reduction in bacterial burden after 1 week, with decreases in TNF-α and IFN-γ levels by 56.6% and 30.2%, respectively. By 2 weeks, bacterial loads decreased by 73.6%, and TNF-α and IFN-γ levels reduced by 53.6% and 67.8%, respectively (Fig. S5). At 4 weeks, 89.2% reduction was observed in pulmonary bacterial counts (Fig. 5C), accompanied by BALF TNF-α and IFN-γ levels of 3.63 and 14.08 pg/mL, corresponding to reductions by 90.9% and 76.1%, respectively (Fig. 5D and E). Histopathological analysis further supported these results. Lung tissues of the infection control group exhibited severe immune cell infiltration, alveolar destruction, and granuloma formation. In contrast, lung tissues of MCCB-04-35- and MCCB-04-37-treated mice showed markedly reduced immune infiltration, preservation of alveolar structures, and a substantial decrease in bacterial burden (Fig. 5F). To determine whether combination therapy could further enhance antibacterial efficacy, mice with chronic *M. avium* infection were treated intraperitoneally for 3 weeks with clarithromycin (CLA, 25 mg/kg), MCCB-04-35 (25 mg/kg), MCCB-04-37 (25 mg/kg), or their combinations (CLA + MCCB-04-35 and CLA + MCCB-04-37, both at 25 mg/kg). No significant toxicity was observed during the treatment period, as indicated by stable body weights and normal clinical appearance. In single-treatment groups, pulmonary bacterial burdens were reduced by approximately 1.4 $\log_{10}$ CFU in the CLA group, 1.7 $\log_{10}$ CFU in the MCCB-04-35 group, and 1.5 $\log_{10}$ CFU in the MCCB-04-37 group relative to the infection control. Combination therapy produced further reductions, with an additional 1.2 $\log_{10}$ CFU decrease for MCCB-04-35 + CLA (*P* < 0.01) and an additional 1.5 $\log_{10}$ CFU decrease for MCCB-04-37 + CLA (*P* < 0.001) compared with the respective monotherapies (Fig. S6). These results demonstrate that the enhanced antibacterial effects observed *in vitro* were recapitulated *in*

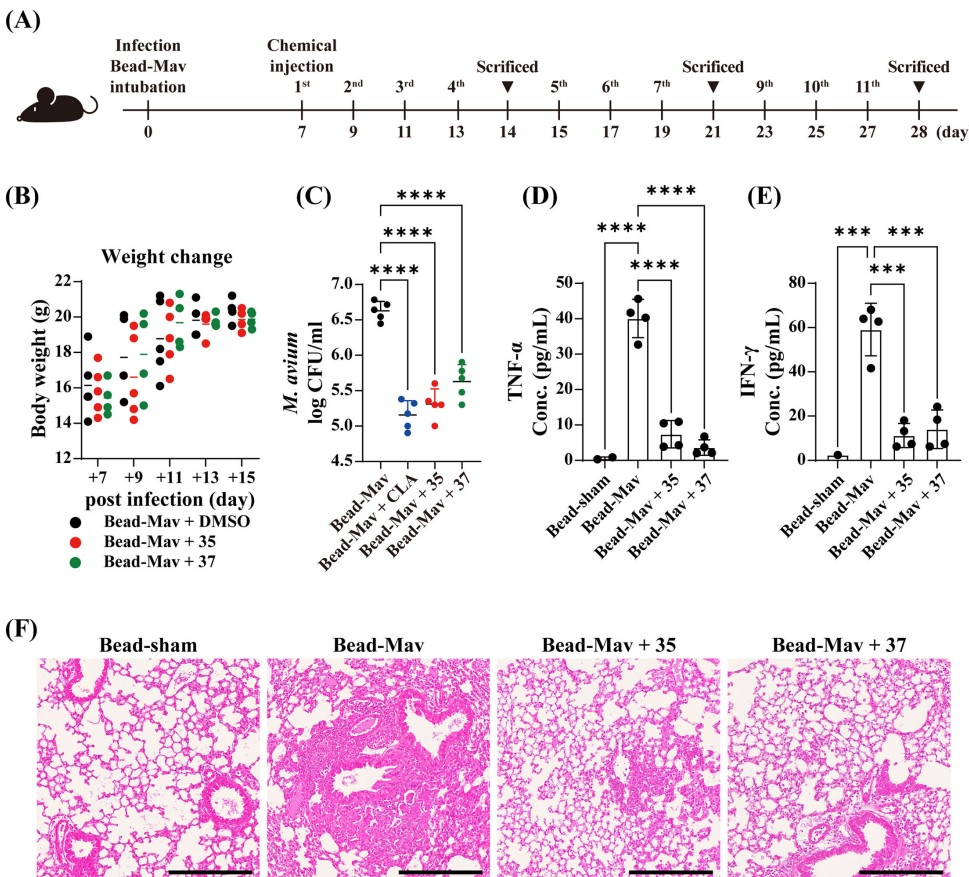

**FIG 5** *In vivo* efficacy of MCCB-04-35 and MCCB-04-37 in a *Mycobacterium avium* bead-induced chronic infection model. (A) Schematic of the experimental design. Mice were infected with *M. avium*-loaded beads (Bead-Mav) via intubation. MCCB-04-35 and MCCB-04-37 were intraperitoneally administered at designated time points. Mice were euthanized at weekly intervals (weeks 2, 3, and 4) post-infection to evaluate treatment efficacy. (B) Changes in body weights of mice. Data points represent individual mice, with weights measured at regular intervals. (C) Bacterial burden in lung tissue. *M. avium* load ($\log_{10}$ CFU/mL) in lung tissue at 4 weeks after infection. (D) TNF-α levels in lung BALF. (E) IFN-γ levels in lung BALF. (F) Hematoxylin and eosin staining of mouse lung tissues. Scale bar = 300 µm. Data points represent individual mice. ***$P < 0.001$, and ****$P < 0.0001$.

*vivo*, suggesting a complementary antibacterial interaction between clarithromycin and MCCB compounds.

## DISCUSSION

This study identified MCCB-04-35 and MCCB-04-37 as promising therapeutic candidates against *M. avium* infection. Both compounds demonstrated potent bacteriostatic activity *in vitro*, in *M. avium*-infected macrophages, and in mice with chronic lung infection. Additionally, transcriptomic analysis provided mechanistic insights into their antimicrobial actions.

Although MICs of MCCB-04-35 (20 µg/mL) and MCCB-04-37 (40 µg/mL) against *M. avium* MHA 104 were higher than that of clarithromycin (0.25 µg/mL), both compounds exhibited strong antimycobacterial effects and significantly reduced intracellular bacterial survival. Marked downregulation of genes involved in oxidative phosphorylation was noticed. This mechanism resembles that of bedaquiline, which targets ATP synthase and inhibits ATP production (19). These findings align with recent efforts prioritizing intracellular activity and mechanism-based screening in developing novel

anti-NTM agents (3, 20, 21), supporting the therapeutic potential of MCCB-04-35 and MCCB-04-37 despite their relatively high MICs.

Additive to synergistic interactions between the MCCB compounds and conventional antibiotics were noticed. Notably, MCCB-04-35 exhibited synergy with ciprofloxacin, and MCCB-04-37 synergized with clarithromycin. Co-treatment resulted in fourfold reductions in MICs, indicating enhanced efficacy. These results are consistent with those of previous reports on synergistic effects among agents targeting bacterial metabolism and intracellular survival (3, 21, 22), highlighting the potential of combination therapy in treating drug-resistant NTM infections.

MCCB-04-35 and MCCB-04-37 also significantly reduced intracellular bacterial burdens in BMDMs without inducing cytotoxicity at concentrations up to 100 µg/mL, suggesting that the observed antimicrobial effects were not attributable to host cell toxicity. *M. avium* evades immune clearance by preventing phagosome–lysosome fusion and resisting oxidative stress (23, 24). Conventional antibiotics, such as β-lactams and aminoglycosides, often exhibit poor membrane permeability and limited activity in acidic phagolysosomal environments, resulting in reduced intracellular efficacy (25). In contrast, MCCB compounds, similar to thiopeptide derivatives like AJ-037 and AJ-206, demonstrated potent intracellular antimicrobial activity and synergism with clarithromycin (26).

Genes associated with nitrogen metabolism and ATP production were downregulated, suggesting metabolic perturbations that may compromise bacterial survival. Suppression of genes involved in L-glutamate biosynthesis, in particular, could reflect interference with nitrogen assimilation and energy metabolism. These metabolic vulnerabilities may explain the enhanced synergy observed with clarithromycin (27, 28). Collectively, these findings indicate that MCCB compounds target core bacterial metabolic pathways, likely via mechanisms distinct from those of classical antimicrobials.

MCCB-04-35 and MCCB-04-37 significantly reduced pulmonary bacterial burden, inflammatory cytokine levels, and histopathological damage in mice with chronic lung infection. Unlike clarithromycin, which has well-documented immunomodulatory activity, and amikacin, whose host-directed effects are inconsistent or minimal, the MCCB compounds effectively reduced immune cell infiltration and preserved alveolar architecture. Clarithromycin, a macrolide antibiotic, is known to suppress NF-κB activation and downregulate proinflammatory cytokines such as TNF-α and IL-6, thereby mitigating inflammation and tissue injury in pulmonary disease models (29–31). In contrast, aminoglycosides such as amikacin have shown variable or context-dependent effects, with reports ranging from anti-inflammatory to negligible host responses depending on the experimental setting (32–34).

In this regard, both MCCB-04-35 and MCCB-04-37 significantly decreased pulmonary TNF-α and IFN-γ levels and alleviated histopathological injury in mice. These findings suggest that the consistent anti-inflammatory profile of MCCB compounds, unlike the variable effects of aminoglycosides, contributes to their dual therapeutic benefits, combining direct antibacterial activity with modulation of infection-associated inflammation. Further studies are warranted to elucidate the precise immunomodulatory mechanisms and clinical relevance.

This study has several limitations. First, the mouse model used may not fully recapitulate the pathophysiology of NTM-induced lung disease in humans. Second, the relatively high MICs highlight the need for formulation optimization or dose refinement. Third, although RNA sequencing provided insights into possible mechanisms of action, proteomic validation is necessary for confirming pathway-level effects. Lastly, pharmacokinetic profiling and assessments of resistance development were not conducted, which are essential for future preclinical development.

In conclusion, MCCB-04-35 and MCCB-04-37 exhibit potent antimicrobial activity against *M. avium* through a multifaceted mechanism involving inhibition of bacterial growth, suppression of intracellular persistence, targeting of metabolic pathways, and

modulation of host inflammation. Their demonstrated efficacies, along with favorable synergy profiles, warrant further investigation towards clinical development.

## MATERIALS AND METHODS

### Strains and growth medium

The reference strains *Mycobacterium abscessus* ATCC 19977 and *M. avium* MAH 104, along with multidrug-resistant clinical isolates (*M. abscessus* 00136-41015, *M. avium* 00136-41015, *Mycobacterium bolletii* 00136-52005, *Mycobacterium fortuitum* 00136-60005, and *M. intracellulare* 00136-43008), were used in this study. All clinical isolates were obtained from the Korean Institute of Tuberculosis (Osong, Republic of Korea). Non-tuberculous mycobacteria were cultured on Middlebrook 7H10 agar (Difco Laboratories, Detroit, MI, USA) supplemented with 10% oleic acid–albumin–dextrose–catalase (OADC; Navi Biotech, Cheonan, Republic of Korea). Liquid cultures were maintained in Middlebrook 7H9 broth (Difco Laboratories) supplemented with 10% OADC and 0.2% glycerol and incubated at 37°C under aerobic conditions.

### Chemical synthesis

Detailed synthetic procedures and spectral data are provided in the Supplemental material ("Synthetic Procedures").

### Cell culture

AML12 cells (CRL-2254; ATCC, Manassas, VA, USA), a murine hepatocyte cell line, were cultured in Dulbecco's modified Eagle's medium/Nutrient Mixture F-12 (Invitrogen, Carlsbad, CA, USA) supplemented with 10% heat-inactivated fetal bovine serum (Invitrogen). Cells were maintained at 37°C in a humidified incubator with 5% $CO_2$ and passaged every 2–3 days upon reaching approximately 80% confluency. Cells were cultured using cell culture plates (SPL Life Sciences, Pocheon, Republic of Korea) under standard conditions.

### Isolation and culture of murine bone marrow-derived macrophages

Bone marrow-derived macrophages (BMDMs) were isolated from the tibias and femurs of 6-week-old C57BL/6 mice and differentiated for 4 days in Dulbecco's modified Eagle's medium supplemented with 10% fetal bovine serum and 25 ng/mL macrophage colony-stimulating factor (R&D Systems, Minneapolis, MN, USA).

### Mice

Wild-type C57BL/6 mice (KOATECH, Pyeongtaek, Republic of Korea), aged 6–7 weeks, were housed under specific pathogen-free conditions and acclimated for 1 week prior to experimentation.

### Cell cytotoxicity assay

AML12 cells ($3 \times 10^3$ cells/well) were seeded into 96-well plates and incubated overnight at 37°C. Cells were then treated with MCCB-04-35 or MCCB-04-37 at different concentrations (1–500 µg/mL) for 24 h. Triton X-100 (0.1%; Sigma-Aldrich, St. Louis, MO, USA) was used as a positive control. After treatment, the culture medium was replaced with fresh medium containing Cell Counting Kit-8 (Dojindo, Japan), and cells were incubated for 1 h. Absorbance at 450 nm was measured using a microplate reader. Cell viability was calculated as a percentage relative to untreated control cells. All experiments were performed in triplicate and independently.

## In vitro anti-NTM activity assay

MICs of MCCB-04-35 and MCCB-04-37 against various NTM strains were determined following the Clinical and Laboratory Standards Institute guidelines M62 and M24 (Wayne, PA, USA). All bacterial strains were grown to logarithmic phase in Middlebrook 7H9 broth supplemented with 10% OADC. Cultures were adjusted to a 0.5 McFarland standard using sterile distilled water and then diluted at 1:100 in 7H9 broth containing 5% OADC. The test compounds were prepared in twofold serial dilutions ranging between 0.625 μg/mL and 320 μg/mL and added to 96-well plates containing the bacterial inoculum. The following controls were included: a bacteria-only control (no drug), a broth-only control (sterility check), and clarithromycin as a positive control. Rapidly growing mycobacteria (*M. abscessus*, *M. fortuitum,* and *M. bolletii*) were incubated at 30°C for 2–5 days, and slow-growing species (*M. avium* and *M. intracellulare*) were incubated at 37°C for 7–14 days. Bacterial growth was assessed by evaluating colony formation. Experiments to determine MICs were performed independently in triplicate.

## Half-maximal inhibitory concentration (IC$_{50}$) determination

The half-maximal inhibitory concentrations (IC$_{50}$) of MCCB-04-35 and MCCB-04-37 against *M. avium* MAH 104 were determined using a CFU-based viability assay. Cultures (5 mL) were grown in Middlebrook 7H9 medium supplemented with 10% OADC at 37°C with shaking at 160 rpm in the presence of serially diluted compounds (0.1–100 μg/mL) from the start of incubation. After 3 days, bacterial suspensions were serially diluted and plated on Middlebrook 7H10 agar for CFU enumeration. Viability was expressed as the percentage of CFU relative to the untreated control, and IC$_{50}$ values were calculated by nonlinear regression using a four-parameter logistic (4PL) model in GraphPad Prism v10.6.1. All assays were performed in triplicate, and results are presented as the mean ± standard deviation (SD).

## Time–kill kinetics assay

*M. avium* MAH 104 was inoculated into Middlebrook 7H9 broth supplemented with 10% OADC at a final density of $1 \times 10^7$ CFU/mL in a 20-μL volume. MCCB-04-35 and MCCB-04-37 were added at final concentrations of 1, 10, 100, 500 μg/mL, based on previously determined MIC values, and cultures were incubated at 37°C for 30 days under shaking condition (160 rpm). At 24-h intervals, cultures were thoroughly mixed, and 100 μL bacterial suspension was serially diluted 10-fold in Dulbecco's phosphate-buffered saline (DPBS; Welgene Inc., Daegu, Republic of Korea). A 10-μL aliquot from each dilution was spot-plated onto Middlebrook 7H10 agar supplemented with 10% OADC. CFUs were enumerated after 7 days of incubation at 37°C. Untreated cultures were considered growth controls. All experiments were independently performed at least twice.

## Checkerboard assay

A checkerboard assay was performed to evaluate synergistic interactions between the MCCB compounds and antibiotics commonly used for *M. avium* infections. Antibiotics were selected based on the Clinical and Laboratory Standards Institute guidelines and included amikacin, ciprofloxacin, moxifloxacin, linezolid, and clarithromycin. Each antibiotic was serially diluted twofold in vertical columns of 96-well plates containing Middlebrook 7H9 broth supplemented with 5% OADC, yielding final concentration ranges of 0.25–16, 0.06–4, 0.12–8, 0.12–8, and 0.01–1 μg/mL for amikacin, ciprofloxacin, moxifloxacin, linezolid, and clarithromycin, respectively. MCCB-04-35 or MCCB-04-37 was diluted in the horizontal rows at final concentrations of 1.5–400 μg/mL. *M. avium* MAH 104 was inoculated at a 1:100 dilution from a 0.5 McFarland suspension in 7H9 medium with 5% OADC, and plates were incubated at 37°C for 7 days. Bacterial growth was

assessed based on colony formation. The fractional inhibitory concentration index (FICI) was calculated using the following formula:

$$FICI = MIC_{A\ in\ combination}/MIC_{A\ alone} + MIC_{B\ in\ combination}/MIC_{B\ alone}.$$

The interaction was interpreted as synergistic (FICI ≤ 0.5), additive (0.5 < FICI ≤ 1), indifferent (1 < FICI ≤ 4), or antagonistic (FICI > 4).

To confirm the synergistic effects observed in the checkerboard assay, CFU counts were determined for the combinations that showed synergy. *M. avium* MAH 104 cultures treated with each single agent or synergistic pair were incubated under the same conditions as above. At the indicated time point, cultures were serially 10-fold diluted in Dulbecco's phosphate-buffered saline (DPBS), and appropriate dilutions were spread on Middlebrook 7H10 agar supplemented with 10% OADC. Plates were incubated at 37°C for 7 days, and visible colonies were enumerated to calculate viable bacterial counts ($\log_{10}$ CFU/mL).

## Intracellular and extracellular bacterial survival assay

BMDMs ($4 \times 10^5$ cells/mL) were seeded in 48-well plates and incubated overnight at 37°C. *M. avium* MAH 104 was added at a multiplicity of infection of 3, followed by a 4-h incubation. Cells were then washed with DPBS to remove extracellular bacteria and treated with or without the test compounds for 24 h. Culture supernatants were collected to assess extracellular bacterial survival. Intracellular bacteria were quantified by lysing cells with 0.1% Triton X-100. Collected medium and cell lysates were serially diluted 10-fold in DPBS and spot-plated onto Middlebrook 7H10 agar supplemented with 10% OADC. CFUs were enumerated after 7 days of incubation at 37°C. Results were expressed as mean CFU/mL ± SD, and assays were performed in triplicate and independently.

## *In vivo* anti-*M. avium* activity assay

*M. avium* MAH 104 was grown on Middlebrook 7H10 agar supplemented with 10% OADC. Single colonies were transferred to Middlebrook 7H9 broth containing 10% OADC and cultured to the logarithmic phase. Cultures were centrifuged, washed with DPBS, and adjusted to an absorbance of 2.0 at 600 nm. To prepare *M. avium*-embedded agar beads, the bacterial suspension was mixed with 7H9 broth containing 2% agar and stirred in mineral oil (Sigma-Aldrich, St. Louis, MO, USA). The resulting agar beads were homogenized, and bacterial concentrations were determined. For infection, C57BL/6 mice were anesthetized and intubated to intratracheally deliver *M. avium*-laden agar beads into the lungs at a final dose of $2 \times 10^7$ CFU per mouse. Mice were allowed to stabilize for 1 week to permit colonization. One week post-infection, the test compounds were intraperitoneally administered at 100 mg/kg every 48 h for 3 weeks. The dose of 100 mg/kg was chosen as the highest non-toxic level to evaluate *in vivo* efficacy in the absence of PK/PD data. The experimental groups included MCCB-04-35, MCCB-04-37, a vehicle control, and a clarithromycin control, with at least five mice per group. To assess bacterial burden, mice were euthanized at weekly intervals, and bronchoalveolar lavage fluid (BALF) and lung tissues were collected. The lungs were homogenized in DPBS, serially diluted, and plated on Middlebrook 7H10 agar for CFU enumeration.

## Combination treatment in the *in vivo* chronic infection model

Mice were infected *via* tracheal intubation with *M. avium*-loaded agar beads as described above. One week after infection, treatment was initiated by intraperitoneal injection every 48 h for 3 weeks. Treatment groups included clarithromycin (CLA, 25 mg/kg), MCCB-04-35 (25 mg/kg), MCCB-04-37 (25 mg/kg), and the corresponding combination groups (CLA + MCCB-04-35 and CLA + MCCB-04-37, both at 25 mg/kg). Body weights and clinical signs were monitored throughout the treatment period to assess potential drug-related toxicity. At 4 weeks post-infection, mice were euthanized, and lungs were aseptically removed and homogenized in PBS. Serial dilutions of the homogenates were

plated on Middlebrook 7H10 agar and incubated at 37°C for 7 days. Bacterial burden was expressed as colony-forming units (CFU) per milliliter of lung homogenate.

## Inflammatory cytokine levels

BALF samples were collected from mice and stored at −80°C until analysis. Levels of inflammatory cytokines, including interleukin-6, tumor necrosis factor alpha (TNF-α), and interferon-gamma (IFN-γ), were measured using uncoated enzyme-linked immunosorbent assay (ELISA) kits (Invitrogen) according to the manufacturers' protocols.

## Histology and immunohistochemistry

Lung tissues were harvested from *M. avium*-infected mice, fixed in 10% neutral-buffered formalin, and embedded in paraffin. Paraffin-embedded tissues were sectioned at 4-µm thickness and stained with hematoxylin and eosin for histopathological evaluation. Inflamed areas were quantified by scanning the entire lung section, and images were analyzed using Motic DSAssistant (Motic VM v.3.0; Motic Asia Corp., Kowloon, Hong Kong). The mean fluorescence intensity of the red threshold was calculated to assess the extent of inflammation.

## Transcriptomic analysis

*Mycobacterium avium* MAH 104 strain was used for transcriptomic analysis. Cultures were grown in Middlebrook 7H9 medium supplemented with 10% OADC at 37°C with agitation at 160 rpm. Cells were inoculated at an initial $OD_{600}$ of 0.08 and treated at the mid-log phase ($OD_{600} \approx 0.6$). Compounds MCCB-04-35 and MCCB-04-37 were prepared as 100 µg/mL stock solutions in DMSO, and cells were exposed to 0.25× MIC (5 µg/mL) for 96 h to minimize bactericidal effects while capturing transcriptional responses. Untreated controls were included. Each condition was performed in three independent biological replicates derived from separate starter cultures. After treatment, cells were harvested and preserved in RNAlater (Thermo Fisher Scientific) to stabilize RNA. The preserved samples were sent to Macrogen Inc. (Seoul, Korea) for RNA extraction, library preparation, and Illumina sequencing, which were performed following the company's standard bulk RNA-sequencing protocol.

## Bulk RNA sequencing

Raw sequencing data underwent quality assessment using FastQC, and low-quality sequences were filtered using Trimmomatic v.0.39 (35). Filtered sequences were aligned to the genome of *M. avium* strain MAH 104 (GCF_000014985.1) utilizing STAR aligner v.2.7.11b (36). Subsequently, mapped BAM files were employed to calculate read counts per sample using featureCounts (37). These counts were normalized and analyzed for differential gene expression using the DESeq2 package (38). Differentially expressed genes (DEGs) were identified based on a *P*-value threshold < 0.05 and an absolute fold change of at least 1.5. Functional enrichment analysis of the DEGs was conducted using FUNAGE-Pro (39).

## Statistical analysis

Statistical analyses were conducted using Prism v.10.3 for Windows (GraphPad Software Inc., San Diego, CA, USA). Unpaired student's *t*-test was used for comparison between two groups, and one-way analysis of variance was used for three or more groups. Data are presented as mean ± SD or s.e. Statistical significance was set at *P* < 0.05.

## ACKNOWLEDGMENTS

This research was supported by a grant of the Korea Health Technology R and D Project through the Korea Health Industry Development Institute (KHIDI), funded by the Ministry of Health & Welfare, Republic of Korea (RS-2024-00336984 and RS-2022-KH130308), and a National Research Foundation of Korea (NRF) grant funded by the Korea government

(MSIT) (RS-2024-00406568 and RS-2025-00515239), supported by a Korea University Grant.

We thank the Korea Basic Science Institute (Ochang) for studies using the Orbitrap Tribrid Fusion Lumos_Ultra High Resolution Multimode Mass Spectrometer (OC104).

Ho-Sung Park: Conceptualization; Data curation; Formal analysis; Validation; Investigation; Visualization; Methodology; Writing—original draft. Do Young Kim: Resources, Investigation, Methodology; Data curation; Formal analysis. Ji Seok Oh: Formal analysis; Validation; Investigation; Visualization; Methodology. Dong Hyon Koo: Formal analysis; Validation; Investigation; Visualization; Methodology. Suyeon Yeom: Formal analysis; Validation; Investigation; Visualization; Methodology. Seungwoo Kim: Formal analysis; Validation; Investigation; Visualization; Methodology. Arjun Gontala: Formal analysis; Validation; Investigation; Visualization; Methodology. Sang-Yeop Lee: Data curation; Formal analysis; Software; Visualization. Dong Ho Kim: Formal analysis; Validation. Kyungho Woo: Formal analysis; validation. Seung Il Kim: Formal analysis; Validation; Software. Jun Young Heo: Formal analysis; Validation. Woosuk Chung: Formal analysis; Validation. Hak Joong Kim: Conceptualization; supervision; Funding acquisition; Investigation; Data curation; Writing-original draft; Writing—review and editing. Chul Hee Choi: Conceptualization; Supervision; Data curation; Funding acquisition; Investigation; Project administration; Writing-original draft; Writing—review and editing.

## AUTHOR AFFILIATIONS

[1]Department of Medical Science, School of Medicine, Chungnam National University, Daejeon, Republic of Korea

[2]Department of Microbiology, School of Medicine, Chungnam National University, Daejeon, Republic of Korea

[3]Translational Immunology Institute, School of Medicine, Chungnam National University, Daejeon, Republic of Korea

[4]System Network Inflammation Control Research Center, Chungnam National University, School of Medicine, Daejeon, Republic of Korea

[5]Department of Chemistry, Korea University, Seoul, Republic of Korea

[6]Korea Basic Science Institute (KBSI), Daejeon, Chungcheongbuk-do, Republic of Korea

[7]Department of Biochemistry, School of Medicine, Chungnam National University, Daejeon, Republic of Korea

[8]Department of Anesthesiology and Pain Medicine, School of Medicine, Chungnam National University, Daejeon, Republic of Korea

[9]Department of Anesthesiology and Pain Medicine, Chungnam National University Hospital, Daejeon, Republic of Korea

## AUTHOR ORCIDs

Ho-Sung Park http://orcid.org/0000-0003-1719-6233
Hak Joong Kim http://orcid.org/0000-0002-3213-2230
Chul Hee Choi http://orcid.org/0000-0003-2373-048X

## AUTHOR CONTRIBUTIONS

Ho-Sung Park, Conceptualization, Data curation, Formal analysis, Investigation, Methodology, Validation, Visualization, Writing – original draft | Do Young Kim, Data curation, Formal analysis, Investigation, Methodology, Resources | Ji Seok Oh, Formal analysis, Investigation, Methodology, Validation, Visualization | Dong Hyon Koo, Formal analysis, Investigation, Methodology, Validation, Visualization | Suyeon Yeom, Formal analysis, Investigation, Methodology, Validation, Visualization | Seungwoo Kim, Formal analysis, Investigation, Methodology, Validation, Visualization | Arjun Gontala, Formal analysis, Investigation, Methodology, Validation, Visualization | Sang-Yeop Lee, Data curation, Formal analysis, Software, Visualization | Dong Ho Kim, Formal analysis, Validation | Kyungho Woo, Formal analysis, Validation | Seung Il Kim, Formal analysis,

Software, Validation | Jun Young Heo, Formal analysis, Validation | Woosuk Chung, Formal analysis, Validation | Hak Joong Kim, Conceptualization, Data curation, Funding acquisition, Investigation, Supervision, Writing – original draft, Writing – review and editing | Chul Hee Choi, Conceptualization, Data curation, Formal analysis, Funding acquisition, Investigation, Methodology, Project administration, Supervision, Writing – original draft, Writing – review and editing

## DATA AVAILABILITY

Full GO enrichment analysis results have been deposited in Figshare at https://doi.org/ 10.6084/m9.figshare.30426742.v2.

## ETHICS APPROVAL

All animal procedures were conducted in accordance with the ethical guidelines of the Chungnam National University School of Medicine and approved by the Institutional Animal Care and Use Committee (IACUC) (approval number: 202,306A-CNU-088, Daejeon, Republic of Korea) and Ministry of Food and Drug Safety of Korea.

## ADDITIONAL FILES

The following material is available online.

### Supplemental Material

**Supplemental figures and tables (Spectrum02160-25-S0001.docx).** Figures S1 to S6 and Tables S1 and S2.
**Supplemental material (Spectrum02160-25-S0002.docx).** Supplemental Materials and Methods.

### Open Peer Review

**PEER REVIEW HISTORY (review-history.pdf).** An accounting of the reviewer comments and feedback.

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
