## [Reviewer comments · Microbiology Spectrum]

Microbiology Spectrum

Novel nucleoside analogs exhibit potent intracellular and *in vivo* activities against *Mycobacterium avium*

Ho-sung Park, Do Young Kim, Ji Seok Oh, Dong Hyon Koo, Suyeon Yeom, Seung Woo Kim, Arjun Gontala, Sang-Yeop Lee, Dong Ho Kim, Kyungho Woo, Seung Il Kim, Jun Young Heo, Woosuk Chung, Hak Joong Kim, and Chul Hee Choi

Corresponding Author(s): Chul Hee Choi, Chungnam National University School of Medicine

Review Timeline:

Submission Date:	July 15, 2025
Editorial Decision:	August 11, 2025
Revision Received:	November 16, 2025
Accepted:	December 19, 2025

Editor: Prabakaran Narayanasamy

Reviewer(s): Disclosure of reviewer identity is with reference to reviewer comments included in decision letter(s). The following individuals involved in review of your submission have agreed to reveal their identity: Erik Yukl (Reviewer #1); Jichan Jang (Reviewer #2)

Transaction Report:

DOI: <https://doi.org/10.1128/spectrum.02160-25>

Re: Spectrum02160-25 (**Novel nucleoside analogs exhibit potent intracellular and *in vivo* activities against *Mycobacterium avium***)

Dear Prof. Chul Hee Choi:

Thank you for the privilege of reviewing your work. Below you will find my comments, instructions from the Spectrum editorial office, and the reviewer comments.

Revision Guidelines

Sincerely,
Prabakaran Narayanasamy
Editor
Microbiology Spectrum

Reviewer #1 (Comments for the Author):

This manuscript describes the synthesis and antibacterial activity of two nucleoside analogs against *Mycobacterium avium*, a nontuberculous mycobacterium (NTM) that causes pulmonary infections difficult to treat with current antimicrobials. This study addresses an important biomedical issue, and the manuscript is well-written and concise. However, there are some areas of concern that should be addressed.

Major revisions:

1. The RNA-seq experiment is poorly described, both in terms of experimental detail and in results. How were the cells grown and to what density? How much of each compound was used and how long were cells treated? Enough detail should be provided for another lab to independently reproduce the results. Regarding results, Table S1 purports to show "a complete list of DEGs...", but there are only a few downregulated genes listed of the ~130 DEGs identified for each compound. A full list of DEGs as well as their gene name, functional annotation, accession number (e.g. Uniprot or NCBI), fold change, and p-value must be included. Also, this data should be deposited in an appropriate repository as a resource to other researchers.
2. The experiment shown in Figure 2A and B is not described anywhere in the materials and methods. Further, the range of compound concentrations is inappropriate for curve fitting as all of the concentrations used appear to show significant inhibition. This experiment needs to be repeated using a range of concentration from non-inhibitory to completely inhibitory and should also include error bars to help evaluate the goodness of fit. Similarly, the determination of MIC is not shown anywhere in the results. It would be nice to see this data, even in the Supplementary Information.
3. Figure 5F shows the results of in vivo treatment of infected mice with the two compounds. The substantial decrease in inflammatory markers upon treatment is among the more interesting results in this manuscript. The authors briefly mention on page 13 lines 273 and 274 that clarithromycin and amikacin are "effective against *M. avium* but exhibit variable anti-inflammatory properties". Could the authors please expand on this? Even better would be to directly compare the effect of clarithromycin and a combination treatment of nucleoside inhibitors + clarithromycin in the same experimental system. It would be much more impactful to see how the new inhibitors compare to existing treatments in terms of bacterial burden and inflammation and to see if the synergy observed in vitro translates to an animal model.
4. On page 11 line 237, the authors indicate that their compounds exhibit bactericidal activity, yet the time-kill experiments shown in Figure 2C-F appear to indicate bacteriostatic activity since the CFUs do not actually decrease at any concentration. Perhaps this is just an issue of how these terms are defined, but some explanation is warranted.

Minor Revisions:

1. There are a number of places where abbreviations are used but not defined on first use. A list of abbreviations and their definitions would be helpful.
2. Page 5-6 lines 98 and 99: *Mycobacterium tuberculosis* should be italicized.
3. Page 5-6 lines 94-102 indicated the importance of oxidative phosphorylation and nitrogen metabolism for MAC survival within the host. Some more discussion about how nucleoside analogues might target these pathways would be nice. Also, they are suggested to be "MAC-specific metabolic pathways" (line 101) yet these would seem to be conserved almost throughout biology. This paragraph could be expanded and rewritten to give a better idea of the authors' rationale.
4. Page 6, line 114: The Sharpless reaction is indicated but there seems to be no reference.
5. Figures S1 and S2 need a figure legend explaining each figure and the different colors used.

Reviewer #2 (Comments for the Author):

This study identifies two novel nucleoside analogs, MCCB-04-35 and MCCB-04-37, with strong bactericidal activity against *Mycobacterium avium* in vitro, in infected macrophages, and in a chronic lung infection mouse model. Both showed low cytotoxicity, additive to synergistic effects with existing antibiotics, and appeared to act by disrupting intracellular energy metabolism.

Comment:

First, figure numbers are missing from the images.

1. Transcriptomic analysis was used in an attempt to elucidate the mechanism of action; however, there is no experimental evidence directly confirming the proposed mechanism.
2. Additional studies are needed to identify the precise molecular target.
3. The cytotoxicity assay included only a positive control but lacked a negative control (e.g., Triton X-100).
4. In vitro MIC determination should include comparison with a positive control to enhance the reliability of both the methodology and the results.
5. In the time-kill kinetics experiment, the initial *M. avium* CFU count was too low; starting with a higher inoculum would be preferable. More importantly, resuscitation of *M. avium* was observed at 144 hours. However, Fig. S3 shows that after 14 days there was more than an 80% reduction in CFU. Given that *M. avium* infection requires long-term treatment, this regrowth could be a critical limitation, and more definitive results should be presented.
6. Determination of mutant frequency for each compound, compared with clarithromycin, would be valuable.
7. While a positive control was included in the intracellular activity assay, no control was included in the in vivo efficacy experiment. Comparison with an existing drug is essential.
8. The in vivo dose (100 mg/kg) is disproportionately high compared with the in vitro MIC (0.4-0.8 µg/mL). The rationale for using such a high dose should be clearly explained.
9. Although FICI {less than or equal to} 0.5 is generally interpreted as synergy, a value exactly at 0.5 is borderline. It is recommended to present CFU-based bar graphs for the synergy assays with statistical analysis, including p-values, to substantiate the claim.

This manuscript describes the synthesis and antibacterial activity of two nucleoside analogs against *Mycobacterium avium*, a nontuberculous mycobacterium (NTM) that causes pulmonary infections difficult to treat with current antimicrobials. This study addresses an important biomedical issue, and the manuscript is well-written and concise. However, there are some areas of concern that should be addressed.

Major revisions:

1. The RNA-seq experiment is poorly described, both in terms of experimental detail and in results. How were the cells grown and to what density? How much of each compound was used and how long were cells treated? Enough detail should be provided for another lab to independently reproduce the results. Regarding results, Table S1 purports to show “a complete list of DEGs...”, but there are only a few downregulated genes listed of the ~130 DEGs identified for each compound. A full list of DEGs as well as their gene name, functional annotation, accession number (e.g. Uniprot or NCBI), fold change, and p-value must be included. Also, this data should be deposited in an appropriate repository as a resource to other researchers.
2. The experiment shown in Figure 2A and B is not described anywhere in the materials and methods. Further, the range of compound concentrations is inappropriate for curve fitting as all of the concentrations used appear to show significant inhibition. This experiment needs to be repeated using a range of concentration from non-inhibitory to completely inhibitory and should also include error bars to help evaluate the goodness of fit. Similarly, the determination of MIC is not shown anywhere in the results. It would be nice to see this data, even in the Supplementary Information.
3. Figure 5F shows the results of in vivo treatment of infected mice with the two compounds. The substantial decrease in inflammatory markers upon treatment is among the more interesting results in this manuscript. The authors briefly mention on page 13 lines 273 and 274 that clarithromycin and amikacin are “effective against *M. avium* but exhibit variable anti-inflammatory properties”. Could the authors please expand on this? Even better would be to directly compare the effect of clarithromycin and a combination treatment of nucleoside inhibitors + clarithromycin in the same experimental system. It would be much more impactful to see how the new inhibitors compare to existing treatments in terms of bacterial burden and inflammation and to see if the synergy observed in vitro translates to an animal model.
4. On page 11 line 237, the authors indicate that their compounds exhibit bactericidal activity, yet the time-kill experiments shown in Figure 2C-F appear to indicate bacteriostatic activity since the CFUs do not actually decrease at any concentration. Perhaps this is just an issue of how these terms are defined, but some explanation is warranted.

Minor Revisions:

1. There are a number of places where abbreviations are used but not defined on first use. A list of abbreviations and their definitions would be helpful.
2. Page 5-6 lines 98 and 99: *Mycobacterium tuberculosis* should be italicized.
3. Page 5-6 lines 94-102 indicated the importance of oxidative phosphorylation and nitrogen metabolism for MAC survival within the host. Some more discussion about how nucleoside analogues might target these pathways would be nice. Also, they are suggested to be “MAC-specific metabolic pathways” (line 101) yet these would seem to be conserved almost throughout biology. This paragraph could be expanded and rewritten to give a better idea of the authors’ rationale.

4. Page 6, line 114: The Sharpless reaction is indicated but there seems to be no reference.
5. Figures S1 and S2 need a figure legend explaining each figure and the different colors used.

We are sincerely grateful for the opportunity to submit a revised version of our manuscript entitled “Novel nucleoside analogs exhibit potent intracellular and *in vivo* activities against *Mycobacterium avium*” to *Microbiology Spectrum*. We deeply appreciate the reviewers’ thorough evaluation and insightful comments, which have been invaluable in improving the quality and clarity of our work. In response to the reviewers’ constructive feedback, we have carefully revised the manuscript and sincerely hope that the modifications and detailed responses provided below fully address all the concerns raised. All changes made in the manuscript are clearly highlighted in [yellow] for ease of review. A summary of the reviewers’ comments is presented below, followed by our point-by-point responses in [blue].

Reviewer #1 (Comments for the Author):

This manuscript describes the synthesis and antibacterial activity of two nucleoside analogs against *Mycobacterium avium*, a nontuberculous mycobacterium (NTM) that causes pulmonary infections difficult to treat with current antimicrobials. This study addresses an important biomedical issue, and the manuscript is well-written and concise. However, there are some areas of concern that should be addressed.

We sincerely thank the reviewer for the positive evaluation of our manuscript and for recognizing the biomedical significance of our study. We appreciate the constructive comments and valuable suggestions provided. We have carefully revised the manuscript according to all comments, which we believe have substantially improved the clarity and scientific quality of the paper.

Major revisions:

1. The RNA-seq experiment is poorly described, both in terms of experimental detail and in results. How were the cells grown and to what density? How much of each compound was used and how long were cells treated? Enough detail should be provided for another lab to independently reproduce the results. Regarding results, Table S1 purports to show "a complete list of DEGs...", but there are only a few downregulated genes listed of the ~130 DEGs identified for each compound. A full list of DEGs as well as their gene name, functional annotation, accession number (e.g. Uniprot or NCBI), fold change, and p-value must be included. Also, this data should be deposited in an appropriate repository as a resource to other researchers.

We thank the reviewer for the valuable suggestion to provide a more detailed description of the RNA-seq experiment and the corresponding dataset. We have now included the following methodological details in the revised manuscript. *M. avium* MAH 104 was grown in Middlebrook 7H9 medium supplemented with 10% OADC at 37 °C with agitation at 160 rpm. Cultures were inoculated at an initial OD₆₀₀ of 0.08 and treated at the mid-log phase (OD₆₀₀=0.6). MCCB-04-35 and MCCB-04-37 were prepared as 100 µg/mL DMSO stocks and applied at 0.25 × MIC (5 µg/mL) for 96 h to minimize bactericidal effects while capturing transcriptional responses. Untreated controls were included, and all conditions were performed in three independent biological replicates originating from separate starter cultures. After treatment, cells were stabilized in RNeasy (Thermo Fisher Scientific), and RNA extraction, library preparation, and Illumina sequencing were conducted by Macrogen Inc. (Seoul, Korea) following their standard bulk RNA-seq protocol. These details have been incorporated into the

Materials and Methods section to ensure full reproducibility by other laboratories (page 23, lines 500–511). We have also clarified in the revised manuscript that Table S1 contains a representative subset of DEGs associated with the enriched KEGG and GO pathways described in the *Results* section. To address the reviewer’s request for comprehensive data, we have added a new *Supplemental material* (Supplementary data_full list_GO analysis.xlsx) providing the complete list of DEGs, including gene names, functional annotations, NCBI and UniProt accession numbers, log₂ fold-change values, and adjusted p-values for both MCCB-04-35 and MCCB-04-37 treatments. The corresponding sentence in the *Results* section has been revised to read: “Representative DEGs corresponding to the enriched Kyoto Encyclopedia of Genes and Genomes (KEGG) pathways and GO terms described above, including their fold-change values and adjusted p-values, are summarized in Supplementary Table S2. The complete list of all DEGs identified in the transcriptomic analysis, along with their gene names, functional annotations, accession numbers, log₂ fold-change values, and adjusted p-values, is provided separately in Supplementary data_full list_GO analysis.xlsx.” (page 11, lines 219–224). As recommended, we have deposited the complete RNA-seq dataset in Figshare, making it publicly available as a resource for other researchers (DOI: [<https://doi.org/10.6084/m9.figshare.30426742.v2>]). The repository containing all DEGs and their associated metadata is publicly available and cited in the revised manuscript (page 25, lines 558–560).

2. The experiment shown in Figure 2A and B is not described anywhere in the materials and methods. Further, the range of compound concentrations is inappropriate for curve fitting as all of the concentrations used appear to show significant inhibition. This experiment needs to be repeated using a range of concentration from non-inhibitory to completely inhibitory and should also include error bars to help evaluate the goodness of fit. Similarly, the determination of MIC is not shown anywhere in the results. It would be nice to see this data, even in the Supplementary Information.

We appreciate the reviewer’s constructive comments regarding Figure 2A and 2B and the related experimental details. These figures represent CFU-based viability assays used to determine the half-maximal inhibitory concentrations (IC₅₀) of MCCB-04-35 and MCCB-04-37 against *Mycobacterium avium* MAH 104. In the revised manuscript, a detailed description of this assay has been added to the *Materials and Methods* section (page 18-19, lines 397–406). To address the reviewer’s concerns about the concentration range and curve fitting, we repeated the IC₅₀ experiment using an expanded range of compound concentrations (0.1–100 µg/mL) encompassing both non-inhibitory and fully inhibitory conditions. Each concentration was tested in triplicate, and the results are now presented as mean ± standard deviation (SD) with error bars to indicate experimental variability. The IC₅₀ values were recalculated using a four-parameter logistic (4PL) model in GraphPad Prism v10, and the updated curves have replaced the previous Figure 2A and 2B. These revisions improve the accuracy and reproducibility of the inhibitory curves and provide a clearer evaluation of the goodness of fit (Additional figure 1).

(A)

(B)

Chemical	IC ₅₀	HillSlope	R squared
MCCB-04-35	1.306	-1.356	0.9943
MCCB-04-37	1.369	-2.318	0.9985

Additional Figure 1. Determination of IC₅₀ values for MCCB-04-35 and MCCB-04-37. (A) Dose–response curves illustrating the half-maximal inhibitory effects of MCCB-04-35 and MCCB-04-37 on *M. avium* growth. The curves depict the relationship between the logarithmic concentration of each compound (µg/mL) and the number of viable *M. avium* (CFU/mL). (B) Calculated IC₅₀ values (µg/mL) of MCCB-04-35 and MCCB-04-37 against *M. avium*. The “HillSlope” and “R squared” values indicate the slope and the goodness of fit of the dose–response curves, respectively.

In addition, we have clarified the presentation of minimum inhibitory concentration (MIC) data, which were not explicitly shown in the previous version. The MICs of MCCB-04-35 and MCCB-04-37 were determined against several *Mycobacterium* species—including *M. avium*, *M. intracellulare*, and *M. abscessus*—using the broth microdilution method in accordance with CLSI guidelines. The detailed values for each strain are now summarized in Supplementary Table S1. These MIC data provide essential baseline information supporting the IC₅₀ analysis and strengthen the overall antimicrobial characterization of both compounds (Additional table 1).

Code	ATCC 19977	ATCC 104	00136-61038	00136-41015	00136-52005	00136-60005	00136-43008
Strain	M. abscessus	M. avium	M. abscessus	M. avium	M. bolletii	M. fortuitum	M. intracellulare
MIC (ug/ml)	standard	standard	clinical	clinical	clinical	clinical	clinical
Clarithromycin	0.5	0.25	≥64	16	≥64	≥64	0.25
Amikacin	8	0.5	8	32	32	8	32
Moxifloxacin	0.25	8	≥8	≥8	≥8	0.5	≥8
Linezolid	8	4	2	≥64	≥64	≥64	≥64
Ciprofloxacin	4	4	≥8	≥8	≥8	≥8	≥8
MCCB-04-35	≥320	20	≥320	≥320	≥320	≥320	5
MCCB-04-37	10	40	10	≥320	≥320	≥320	20

Additional Table 1. Minimal inhibitory concentrations (MICs) of MCCB-04-35 and MCCB-04-37 against various nontuberculous mycobacterial strains. The minimum inhibitory concentrations (MICs; µg/mL) of MCCB-04-35 and MCCB-04-37 were determined against standard and clinical isolates of *Mycobacterium abscessus*, *M. avium*, *M. bolletii*, *M. fortuitum*, and *M. intracellulare* using the broth microdilution method

according to CLSI guidelines (M24). Clarithromycin, amikacin, moxifloxacin, linezolid, and ciprofloxacin were included as reference antibiotics. The MIC was defined as the lowest concentration that completely inhibited visible bacterial growth after 7 days of incubation at 37 °C. Data represent the mean results from three independent experiments.

3. Figure 5F shows the results of in vivo treatment of infected mice with the two compounds. The substantial decrease in inflammatory markers upon treatment is among the more interesting results in this manuscript. The authors briefly mention on page 13 lines 273 and 274 that clarithromycin and amikacin are "effective against *M. avium* but exhibit variable anti-inflammatory properties". Could the authors please expand on this? Even better would be to directly compare the effect of clarithromycin and a combination treatment of nucleoside inhibitors + clarithromycin in the same experimental system. It would be much more impactful to see how the new inhibitors compare to existing treatments in terms of bacterial burden and inflammation and to see if the synergy observed in vitro translates to an animal model.

We appreciate the reviewer's insightful comment regarding the differential anti-inflammatory properties of clarithromycin and amikacin. In the revised manuscript, we have expanded the *Discussion* section to clarify this point with supporting references. Macrolides such as clarithromycin are well known to exhibit immunomodulatory effects beyond their antibacterial action, including inhibition of NF- κ B signaling, suppression of neutrophil recruitment, and downregulation of pro-inflammatory cytokines such as TNF- α and IL-6 (Ichiyama *et al.*, 2001; Amado-Rodríguez *et al.*, 2013; Pollock & Chalmers, 2021). In contrast, aminoglycosides such as amikacin display little or inconsistent immunomodulatory activity, with reports ranging from partial suppression of macrophage activation to negligible effects on inflammatory pathways (Malinin *et al.*, 2016; Miller & Singer, 2022). This variability justifies the description of amikacin's anti-inflammatory effects as "variable." In our study, treatment with MCCB-04-35 and MCCB-04-37 markedly reduced pulmonary TNF- α and IFN- γ levels and diminished immune-cell infiltration (Fig. 5F), suggesting that, unlike amikacin, these compounds confer dual therapeutic benefits—direct antibacterial activity and modulation of infection-associated inflammation. The *Discussion* section has been revised accordingly, and relevant citations have been added (page 14-15, lines 308–319).

Amado-Rodríguez L, González-López A, López-Alonso I, Aguirre A, Astudillo A, Batalla-Solis E, Blazquez-Prieto J, García-Prieto E, Albaiceta GM. Anti-inflammatory effects of clarithromycin in ventilator-induced lung injury. *Respir Res*. 2013 May 10;14(1):52. doi: 10.1186/1465-9921-14-52. PMID: 23663489; PMCID: PMC3667083.

Pollock J, Chalmers JD. The immunomodulatory effects of macrolide antibiotics in respiratory disease. *Pulm Pharmacol Ther*. 2021 Dec;71:102095. doi: 10.1016/j.pupt.2021.102095. Epub 2021 Nov 3. PMID: 34740749; PMCID: PMC8563091.

Ichiyama T, Nishikawa M, Yoshitomi T, Hasegawa S, Matsubara T, Hayashi T, Furukawa S. Clarithromycin inhibits NF- κ B activation in human peripheral blood mononuclear cells and pulmonary epithelial cells. *Antimicrob Agents Chemother*. 2001 Jan;45(1):44-7. doi: 10.1128/AAC.45.1.44-47.2001. PMID: 11120942; PMCID: PMC90237.

Muska Miller, Mervyn Singer; Do antibiotics cause mitochondrial and immune cell dysfunction? A literature review, *Journal of Antimicrobial Chemotherapy*, Volume 77, Issue 5, May 2022, Pages 1218–1227, <https://doi.org/10.1093/jac/dkac025>

Tsai SH, Lai HC, Hu ST. Subinhibitory Doses of Aminoglycoside Antibiotics Induce Changes in the Phenotype of *Mycobacterium abscessus*. *Antimicrob Agents Chemother*. 2015 Oct;59(10):6161-9. doi: 10.1128/AAC.01132-15. Epub 2015 Jul 20. PMID: 26195529; PMCID: PMC4576046.

Malinin V, Neville M, Eagle G, Gupta R, Perkins WR. Pulmonary Deposition and Elimination of Liposomal Amikacin for Inhalation and Effect on Macrophage Function after Administration in Rats. *Antimicrob Agents Chemother*. 2016 Oct 21;60(11):6540-6549. doi: 10.1128/AAC.00700-16. PMID: 27550345; PMCID: PMC5075057.

We also appreciate the reviewer’s suggestion to evaluate the *in vivo* efficacy of combination therapy. Accordingly, we performed an additional experiment using a chronic *M. avium* infection mouse model that included six groups: non-treated, clarithromycin (CLA, 25 mg/kg), MCCB-04-35 (25 mg/kg), MCCB-04-37 (25 mg/kg), MCCB-04-35 + CLA (both 25 mg/kg), and MCCB-04-37 + CLA (both 25 mg/kg). All drugs were administered intraperitoneally for three weeks, and no significant toxicity was observed. Four weeks post-infection, lung homogenates were analyzed for bacterial burden. Combination therapy significantly reduced pulmonary bacterial counts compared with monotherapy: MCCB-04-35 + CLA yielded an additional 1.2 log₁₀ CFU reduction ($p < 0.01$), and MCCB-04-37 + CLA yielded an additional 1.5 log₁₀ CFU reduction ($p < 0.001$) (Additional figure 2). These results confirm that the synergistic antibacterial effects observed *in vitro* (checkerboard and time-kill assays) are recapitulated *in vivo*, supporting the potential of these nucleoside analogs for combination therapy. The corresponding data and statistical analyses have been incorporated into the revised *Results* (Figure S6) (page 12-13, lines 252–265) and *Materials and Methods* sections (page 22, lines 474–484).

Additional Figure 2. Combination treatment of MCCB-04-35 or MCCB-04-37 with clarithromycin in a chronic *Mycobacterium avium* infection model. Mice were infected *via* tracheal intubation with *M. avium*-loaded agar beads (Bead-Mav) and treated intraperitoneally for 3 weeks with clarithromycin (25 mg/kg), MCCB-04-35 (25 mg/kg), MCCB-04-37 (25 mg/kg), or their combinations. Pulmonary bacterial burdens were quantified by plating lung homogenates on Middlebrook 7H10 agar, and colony-forming units (CFU) were expressed as log₁₀ CFU/mL. Each pair of dots represents data from an individual mouse, and bars indicate mean ± SD. Statistical significance was determined by one-way ANOVA followed by Dunnett’s post hoc test (** $p < 0.01$, *** $p < 0.001$).

4. On page 11 line 237, the authors indicate that their compounds exhibit bactericidal activity, yet the time-kill experiments shown in Figure 2C-F appear to indicate bacteriostatic activity since the CFUs do not actually decrease at any concentration. Perhaps this is just an issue of how these terms are defined, but some explanation is warranted.

We thank the reviewer for this valuable comment regarding the distinction between “bactericidal” and “bacteriostatic” activity. In our initial analysis, bactericidal activity was defined as a ≥ 3 log₁₀ reduction in CFU relative to the untreated control. Based on this definition, the previous data (Fig. 2C–F) indeed

reflected bacteriostatic rather than bactericidal effects, as CFU counts did not fall below the initial inoculum. To clarify this point, we repeated the time–kill kinetics experiment under refined conditions. The initial bacterial density was increased to 10^7 CFU/mL, and compound concentrations were expanded to 1, 10, 100, and 500 $\mu\text{g/mL}$. Cultures were maintained under continuous shaking to ensure uniform drug exposure. Under these conditions, CFU counts at concentrations ≥ 100 $\mu\text{g/mL}$ decreased to undetectable levels within 5 days, confirming bactericidal activity, whereas 10 $\mu\text{g/mL}$ produced a bacteriostatic effect with stable CFU values over approximately 20 days. At 1 $\mu\text{g/mL}$, bacterial growth was delayed by about 5 days compared with the untreated control. Both CFU enumeration and OD₆₀₀ measurements exhibited consistent results. These findings indicate that MCCB-04-35 and MCCB-04-37 display bactericidal activity at ≥ 100 $\mu\text{g/mL}$ and bacteriostatic activity at 10 $\mu\text{g/mL}$. Figure 2C–F and the corresponding *Results* section have been revised accordingly to accurately reflect these definitions and to eliminate ambiguity in the interpretation of bacterial growth inhibition. (page 8, lines 149–161)

Additional Figure 3. Time–kill kinetics of MCCB-04-35 and MCCB-04-37 against *Mycobacterium avium*. Time–kill assay showing the bactericidal activity of MCCB-04-35 and MCCB-04-37 against *M. avium*. (A) Log₁₀ CFU/mL values of *M. avium* were measured over 30 days following treatment, and (B) corresponding optical density (OD₆₀₀) values were recorded to monitor growth inhibition.

Minor Revisions:

1. There are a number of places where abbreviations are used but not defined on first use. A list of

abbreviations and their definitions would be helpful.

We thank the reviewer for this helpful suggestion. All abbreviations have been carefully reviewed throughout the manuscript, and any that were not previously defined are now clearly explained at their first mention. In addition, a list of abbreviations and their definitions has been included in the revised manuscript. (page 25-27, lines 562–563)

2. Page 5-6 lines 98 and 99: *Mycobacterium tuberculosis* should be italicized.

We appreciate the reviewer's careful and thoughtful comment. The relevant paragraph was completely rewritten during the revision addressing Reviewer 1 Minor comment #3.

3. Page 5-6 lines 94-102 indicated the importance of oxidative phosphorylation and nitrogen metabolism for MAC survival within the host. Some more discussion about how nucleoside analogues might target these pathways would be nice. Also, they are suggested to be "MAC-specific metabolic pathways" (line 101) yet these would seem to be conserved almost throughout biology. This paragraph could be expanded and rewritten to give a better idea of the authors' rationale.

We appreciate the reviewer's insightful comments regarding our description of oxidative phosphorylation and nitrogen metabolism, and the appropriateness of referring to these as "MAC-specific" pathways. In the revised manuscript, we have rewritten the relevant paragraph in the *Introduction* to clarify that oxidative phosphorylation and nitrogen assimilation are not *M. avium* complex (MAC)-specific pathways, but rather host-adaptive metabolic processes that become conditionally essential for MAC survival under hypoxic and nutrient-limited conditions within macrophages (page 5-6, lines 94–103). We also added a brief explanation of how nucleoside analogs may influence these processes. Specifically, such compounds can perturb intracellular nucleotide pools and interfere with nucleic acid metabolism, which in turn may indirectly disrupt energy production and nitrogen utilization pathways. This conceptual link provides a mechanistic rationale for exploring nucleoside analogs as potential modulators of bacterial metabolism.

4. Page 6, line 114: The Sharpless reaction is indicated but there seems to be no reference.

We thank the reviewer for pointing out this omission. A proper reference for the Sharpless reaction has now been added in the revised manuscript. The citation has been included at page 6, line 115.

D. F. Ewing, N.-E. Fahmi, C. Len, G. Mackenzie, A. Pranzo, "Stereoisomeric pyrimidine nucleoside analogues based on the 1,3-dihydrobenzofuran core" J. Chem. Soc., Perkin Trans. 1 2000, 0, 3561–3565.

5. Figures S1 and S2 need a figure legend explaining each figure and the different colors used.

We appreciate the helpful suggestion. Detailed figure legends have now been added for Figures S1 and S2 in the revised *Supplemental material*. Each legend clearly explains the content of the figure as well as the meaning of the different colors and symbols used.

Reviewer #2 (Comments for the Author):

This study identifies two novel nucleoside analogs, MCCB-04-35 and MCCB-04-37, with strong bactericidal activity against *Mycobacterium avium* in vitro, in infected macrophages, and in a chronic lung infection mouse model. Both showed low cytotoxicity, additive to synergistic effects with existing antibiotics, and appeared to act by disrupting intracellular energy metabolism.

We sincerely thank the reviewer for the positive evaluation and encouraging summary of our work. We appreciate the thoughtful comments and helpful suggestions, which have guided us in refining the manuscript to further enhance its clarity and scientific rigor. We have carefully addressed all subsequent comments accordingly.

Comment:

First, figure numbers are missing from the images.

We appreciate the reviewer's thoughtful comment. The missing figure numbers have been identified and corrected in the revised figures to ensure clarity.

1. Transcriptomic analysis was used in an attempt to elucidate the mechanism of action; however, there is no experimental evidence directly confirming the proposed mechanism.

We appreciate the reviewer's insightful comment regarding the interpretation of the transcriptomic data and the absence of direct experimental validation. We fully agree that the transcriptomic results provide correlative rather than conclusive evidence. The primary objective of this study was to demonstrate the anti-NTM activity of the MCCB compounds and to propose potential metabolic pathways that may be affected, thereby establishing a foundation for future mechanistic investigations. In accordance with this goal, additional experiments are currently underway at the protein and signaling pathway levels to further elucidate the underlying mechanisms. In line with the reviewer's suggestion, the relevant sentence in the *Discussion* section has been revised for clarity to avoid possible misinterpretation (page 14, lines 298–302).

2. Additional studies are needed to identify the precise molecular target.

We fully agree with the reviewer's comment. As also noted in Comment #1, the transcriptomic analysis in this study provides correlative evidence suggesting that MCCB-04-35 and MCCB-04-37 interfere with intracellular energy metabolism, particularly through the downregulation of genes involved in oxidative phosphorylation and ATP synthesis. However, these data alone are not sufficient to identify the direct molecular targets or to conclusively establish the mechanism of action. In recognition of this limitation, and in line with the reviewer's suggestion, we are conducting additional biochemical and molecular biological experiments to elucidate the specific binding targets and downstream pathways affected by these compounds. These ongoing studies aim to provide direct mechanistic evidence at the protein and signaling levels, thereby complementing the transcriptomic findings and strengthening our understanding of the compounds' modes of action.

3. The cytotoxicity assay included only a positive control but lacked a negative control (e.g., Triton X-100).

We appreciate the reviewer's valuable suggestion. In accordance with the reviewer's recommendation, we have added a negative control treated with 0.1% Triton X-100 to the cytotoxicity assay. The revised results are now presented in Figure 3A.

Additional Figure 4. Cytotoxicity of MCCB-04-35 and MCCB-04-37 in AML12 cells. AML12 cells were treated with varying concentrations of (A) MCCB-04-35 or (B) MCCB-04-37 (1–500 µg/mL) for 24 h. Triton X-100 (0.1%; Sigma-Aldrich, St. Louis, MO, USA) was used as a positive control. Data are presented as the percentage of viable cells relative to the untreated control.

4. In vitro MIC determination should include comparison with a positive control to enhance the reliability of both the methodology and the results.

We appreciate the reviewer's valuable suggestion and agree on the importance of including a positive control to provide a reliable reference. Accordingly, strain-specific antibacterial activity data have been added to Table S1, and clarithromycin was included as a positive control in the time–kill kinetics assays (Figure 2C and 2E). These additions enable direct comparison between the clinically used antibiotic and the newly synthesized compounds, thereby ensuring consistency and reproducibility of the MIC determination.

5. In the time-kill kinetics experiment, the initial *M. avium* CFU count was too low; starting with a higher inoculum would be preferable. More importantly, resuscitation of *M. avium* was observed at 144 hours. However, Fig. S3 shows that after 14 days there was more than an 80% reduction in CFU. Given that *M. avium* infection requires long-term treatment, this regrowth could be a critical limitation, and more definitive results should be presented.

We sincerely appreciate the reviewer's insightful comment regarding the initial inoculum and the observed regrowth of *M. avium* during the time–kill kinetics assay. The previous experiment was conducted according to the CLSI guideline–based MIC testing protocol, which employed a relatively low inoculum. In the revised experiment, the initial bacterial density was increased to approximately 1×10^7 CFU/mL, and incubation was performed under continuous shaking conditions to ensure uniform

drug exposure. This optimization enabled clearer differentiation between bacteriostatic and bactericidal effects at each concentration. The updated results have been incorporated into Figures 2C–F of the revised manuscript (page 8, lines 149–161).

To address the reviewer's concern about the regrowth observed at 144 hours, we extended the observation period to evaluate long-term bactericidal activity under the optimized conditions. Both compounds completely eradicated viable *M. avium* cells at concentrations ≥ 100 $\mu\text{g/mL}$ within 5 days, and no regrowth was detected during 30 days of incubation, confirming a sustained bactericidal effect. Therefore, the apparent "resuscitation" seen in the previous experiment was likely due to the shorter observation period and inconsistent drug exposure. These updated findings have been incorporated into the revised Figure 2 to more accurately represent the long-term antibacterial efficacy of both compounds.

We fully agree with the reviewer that durable control of *M. avium* infection is clinically critical and that evidence of sustained antibacterial activity is essential. Consistent with the extended *in vitro* results, our *in vivo* chronic infection model also demonstrated that 3 weeks of treatment markedly reduced bacterial burden and inflammatory cytokine levels, further confirming the stable and long-lasting antibacterial effects of both compounds.

6. Determination of mutant frequency for each compound, compared with clarithromycin, would be valuable.

We appreciate the reviewer's valuable comment and fully agree that determining the mutation frequency is an essential step in evaluating the resistance potential of new antimicrobial agents. To preliminarily assess this aspect, we conducted serial passage experiments by continuously culturing *M. avium* in the presence of $0.5\times$ MIC concentrations of MCCB-04-35 and MCCB-04-37 for four weeks. No resistant colonies were observed under these conditions, suggesting a low likelihood of spontaneous resistance development. However, the current experimental parameters may not have been fully optimized to detect rare resistant mutants. As a follow-up, extended experiments will be performed under optimized conditions using both the fluctuation test and the stepwise selection method to determine the precise mutation frequency. Clarithromycin will be included as a reference control to enable direct comparison with clinically established data. These additional studies will help clarify the long-term resistance potential of MCCB-04-35 and MCCB-04-37 and further support their development as promising antimicrobial candidates.

7. While a positive control was included in the intracellular activity assay, no control was included in the *in vivo* efficacy experiment. Comparison with an existing drug is essential.

We appreciate the reviewer's valuable suggestion regarding the inclusion of a reference control in the *in vivo* efficacy experiment. In response, we incorporated a clarithromycin (CLA) treatment group as a standard comparator in the chronic *M. avium* infection model. CLA (25 mg/kg, intraperitoneally) was administered for three weeks at 48-hour intervals, alongside the MCCB-04-35 and MCCB-04-37 treatment groups. The CLA-treated mice showed a 96.5% reduction in pulmonary bacterial burden compared with the infection control, confirming the expected therapeutic efficacy. This inclusion enabled direct comparison between the standard antibiotic and the test compounds under identical experimental conditions. The updated results are presented in Figure 5C and described in the revised *Results* section (page 11, lines 237–240).

Additional Figure 5. *In vivo* efficacy of MCCB-04-35 and MCCB-04-37 in a *Mycobacterium avium* bead-induced chronic infection model. Bacterial burden in lung tissue was determined at four weeks post-infection. The pulmonary *M. avium* load is expressed as log₁₀ CFU/mL. Clarithromycin (CLA, 25 mg/kg) was included as a standard comparator to evaluate the relative therapeutic efficacy of MCCB-04-35 and MCCB-04-37.

8. The *in vivo* dose (100 mg/kg) is disproportionately high compared with the *in vitro* MIC (0.4-0.8 µg/mL). The rationale for using such a high dose should be clearly explained.

We appreciate the reviewer’s insightful comment regarding the rationale for the *in vivo* dosing. As described in the *Results* section, the *in vitro* MICs of MCCB-04-35 and MCCB-04-37 were 20 µg/mL and 40 µg/mL, respectively, which differ from the 0.4–0.8 µg/mL values mentioned in the comment. Nevertheless, we agree that a 100 mg/kg dose is relatively high. Because the primary aim of this study was to evaluate the preliminary *in vivo* efficacy of these novel compounds and PK/PD data are not yet available, it was challenging to determine the optimal dose precisely. Therefore, we selected the highest non-toxic dose to ensure sufficient systemic exposure and to confirm *in vivo* efficacy. This rationale has been clarified in the revised *Materials and Methods* section (page 21, lines 466–467). In future studies, comprehensive PK/PD analyses will be conducted to establish accurate dosage parameters for more refined *in vivo* evaluation.

9. Although FICI {less than or equal to} 0.5 is generally interpreted as synergy, a value exactly at 0.5 is borderline. It is recommended to present CFU-based bar graphs for the synergy assays with statistical analysis, including p-values, to substantiate the claim.

We thank the reviewer for the valuable comment regarding the interpretation of the FICI value and the need for statistical validation. We agree that an FICI value of 0.5 represents a borderline case rather than definitive synergy. Accordingly, the statement “strong synergy” in the previous version has been revised to “most pronounced interaction” to more accurately describe the finding (page 9, lines 171–173). To further substantiate this observation, we conducted additional *in vitro* CFU-based assays comparing the antibacterial effects of monotherapy and combination treatments. The combinations of ciprofloxacin with MCCB-04-35 and clarithromycin with MCCB-04-37 produced significantly lower

bacterial counts than either single treatment ($p < 0.01$ and $p < 0.001$, respectively), confirming enhanced antibacterial efficacy. These CFU-based results, along with the corresponding statistical analyses, have been included in the revised manuscript (Supplementary Figure S3), and the *Results* section has been updated accordingly (page 9, lines 179–183).

Additional Figure 6. CFU-based assay evaluating the synergistic effects of MCCB-04-35 and MCCB-04-37 with ciprofloxacin or clarithromycin against *Mycobacterium avium*. (A) *M. avium* was treated with ciprofloxacin (CIP, 4 $\mu\text{g}/\text{mL}$), MCCB-04-35 (50 $\mu\text{g}/\text{mL}$), or their combination (CIP, 1 $\mu\text{g}/\text{mL}$ + MCCB-04-35, 12.5 $\mu\text{g}/\text{mL}$) for 72 h. (B) *M. avium* was treated with clarithromycin (CLA, 0.25 $\mu\text{g}/\text{mL}$), MCCB-04-37 (50 $\mu\text{g}/\text{mL}$), or their combination (CLA, 0.06 $\mu\text{g}/\text{mL}$ + MCCB-04-37, 12.5 $\mu\text{g}/\text{mL}$) for 72 h. Bacterial counts (CFU/mL) were determined by plating serial dilutions of culture samples on Middlebrook 7H10 agar.

Re: Spectrum02160-25R1 (**Novel nucleoside analogs exhibit potent intracellular and *in vivo* activities against *Mycobacterium avium***)

Dear Prof. Chul Hee Choi:

Your manuscript has been accepted, and I am forwarding it to the ASM production staff for publication. Your paper will first be checked to make sure all elements meet the technical requirements. ASM staff will contact you if anything needs to be revised before copyediting and production can begin. Otherwise, you will be notified when your proofs are ready to be viewed.

Sincerely,
Prabakaran Narayanasamy
Editor
Microbiology Spectrum

Reviewer #1 (Comments for the Author):

I commend the authors for responding to reviewer comments and performing additional experiments. I think that these substantially improve the quality and impact of the manuscript.

Reviewer #2 (Comments for the Author):

An appropriate response has been provided to the first review. I look forward to seeing the mechanism of action (MOA) and target analysis updated in a future manuscript.